# Long-term healthcare provider availability following large-scale hurricanes: A difference-in-differences study

Sue Anne Bell [1,2☯] *, Katarzyna Klasa [3☯], Theodore J. Iwashyna[2,4,5☯], Edward C. Norton[2,3☯], Matthew A. Davis [1,2,6☯]

1 Department of Systems, Populations, and Leadership, University of Michigan School of Nursing, Ann Arbor, Michigan, United States of America, 2 Institute for Healthcare Policy and Innovation, University of Michigan, Ann Arbor, Michigan, United States of America, 3 Department of Health Management and Policy, University of Michigan School of Public Health, Ann Arbor, Michigan, United States of America, 4 Department of Internal Medicine, University of Michigan Medical School, Ann Arbor, Michigan, United States of America, 5 University of Michigan Institute for Social Research, Ann Arbor, Michigan, United States of America, 6 Department of Learning Health Sciences, University of Michigan Medical School, Ann Arbor, Michigan, United States of America

☯ These authors contributed equally to this work.
* sabell@umich.edu

**Data Availability Statement:** The data underlying the results presented in the study are available from the Federal Emergency Management Agency at https://www.fema.gov/data-sets and from the

## Abstract

### Background

Hurricanes Katrina and Sandy were two of the most significant disasters of the 21st century that critically impacted communities and the health of their residents. Despite the assumption that disasters affect access to healthcare, to our knowledge prior studies have not rigorously examined availability of healthcare providers following disasters.

### Objective

The objective of this study was to examine availability of healthcare providers following large-scale hurricanes.

### Methods

Using historical data on healthcare providers from the National Plan and Provider Enumeration System and county-level population characteristics, we conducted a quasi-experimental study to examine the effect of large-scale hurricanes on healthcare provider availability in the short-term and long-term. We separately examined availability of primary care physicians, medical specialists, surgeons, and nurse practitioners. A difference-in-differences analysis was used to control for time variant factors comparing county-level health care provider availability in affected and unaffected counties the year before Hurricanes Katrina and Sandy, to five years after each storm.

National Plan and Provider Enumeration System at https://www.cms.gov/Regulations-and-Guidance/Administrative-Simplification/NationalProvIdentStand/DataDissemination.

**Funding:** Research reported in this publication was supported by National Institute on Aging of the National Institutes of Health (https://www.nia.nih.gov) under award number K23AG059890 (Author: SB). The content is solely the responsibility of the authors and does not necessarily represent the official views of the National Institutes of Health. The funders had no role in study design, data collection and analysis, decision to publish, or preparation of the manuscript.

**Competing interests:** The authors have declared that no competing interests exist.

## Results

Counties affected by Hurricane Katrina compared to unaffected locales experienced a decrease of 3.59 primary care physicians per 10,000 population (95% CI: -6.5, -0.7), medical specialists (decrease of 5.9 providers per 10,000 (95% CI: -11.3, -0.5)), and surgeons (decrease of 2.1 (95% CI: -3.8, -0.37)). However, availability of nurse practitioners did not change appreciably. Counties affected by Hurricane Sandy exhibited less pronounced changes. Changes in availability of primary care physicians, nurse practitioners, medical specialists, and surgeons were not statistically significant.

## Conclusion

Large-scale hurricanes appear to affect availability of healthcare providers for up to several years following impact of the storm. Effects vary depending on the characteristics of the community. Primary care physicians and medical specialists availability was the most impacted, potentially having long-term implications for population health in the context of disaster recovery.

## Introduction

Hurricanes Katrina and Sandy were two of the most significant disasters of the 21st century, critically impacting communities [1–7]. During Hurricane Katrina in 2005, over 80% of New Orleans flooded, 1,577 people died, and the total sustained damages were estimated at $108 billion [8]. Widespread evacuations and long-term migration away from affected locales occurred. Orleans and Jefferson parishes, both in the New Orleans metropolitan statistical area, experienced a 40% and 19% drop in population respectively [9]. Seven years later, in 2012, Hurricane Sandy made landfall on the eastern seaboard of the United State resulting in severe damage to the New York and New Jersey area. Hurricane Sandy caused over $50 billion in damages [10] and resulted in an estimated 72 deaths [11]. Hurricanes Katrina and Sandy were two of the most damaging, and costly, in recent history, but also have now occurred with enough elapsed time in which to evaluate long-term sequelae on communities [12].

Healthcare provider availability has been associated with reduced mortality [13] and access to primary care physicians has been shown to affect health outcomes among Medicare beneficiaries [14]. Maintaining adequate access to healthcare services is critical for community recovery after disaster. As such, there is considerable interest in understanding the extent to which healthcare resources are affected by disasters. Underpinning disaster preparedness planning is the assumption that access to healthcare services diminishes for a period of time (be it short or long) as a community recovers. Furthermore, the socioeconomic characteristics of the population likely contribute to the recovery of the community, which in turn influences the degree to which healthcare provider remain in the locale, as healthcare providers tend to concentrate in affluent locales [15].

However, very few studies have empirically examined the effect of disasters on healthcare provider availability. One such study of physician characteristics and outflow after the 2011 Great Japan Earthquake and Fukishima nuclear accident found that early career physicians were more at risk for leaving the affected area in the year after the earthquake [16]. Other prior studies are descriptive in nature, examining short-term changes after Hurricane Katrina, finding that roughly 25% of the New Orleans area's physicians had returned to practice two years

after Katrina [17]. To our knowledge no prior study has examined long-term availability of health care providers after disasters. Therefore, we rigorously evaluated the impact of hurricanes on healthcare provider availability. To explore how effects may differ dependent upon community-level socioeconomic characteristics, we selected hurricanes Katrina and Sandy which affected two very different populations.

## Methods

Using a combination of provider location and population data, we performed a quasi-experimental study to examine the extent to which healthcare provider availability is affected by large-scale hurricanes up to five years post hurricane. To control for time-variant factors, we compared availability before versus after hurricane landfall using difference-in-differences (DID) analysis. DID uses data from treatment groups and control groups (in our analysis, hurricane-affected counties and unaffected counties, respectively) to obtain an appropriate counterfactual to estimate a causal effect over time [18]. This study used publicly available data; and, therefore, was determined to be exempt from review by the University of Michigan Health Sciences and Behavioral Sciences Institutional Review Board.

### Healthcare provider availability

Historic data from the National Plan and Provider Enumeration System (NPPES) were used to identify healthcare providers. The NPPES provides basic information about individual healthcare providers who have a National Provider Identifier (NPI). We examined availability of four types of healthcare providers: primary care physicians, nurse practitioners, medical specialists, and surgeons, following Centers for Medicare and Medicaid Services (CMS) provider taxonomy and specialty codes [19]. We operationally defined primary care physicians as those specializing in family practice, internal medicine, or general practice. Nurse practitioners provide primary care often without the supervision of a physician, and supplement primary care provision in many communities. This study excluded certain types of advanced practice nurses such as certified midwifes, nurse anesthetists, and certified clinical nurse specialists. To be included in this study, providers had to have an active license and valid primary practice address.

In 2007, the CMS changed the provider identifier used on billed services from Unique Provider Identification Number (UPIN) to NPI. The NPI is a unique 10-digit numeric identification number for covered healthcare providers, and is permanently associated with a specific individual regardless of changes in practice or location [20, 21]. Any provider, organization or individual, that is a HIPAA-covered entity must have an NPI [22]. For this study, only individual healthcare providers were included. NPPES data have been shown to provide reasonably accurate, up-to-date address information for physicians billing public and private insurers, when compared to similar databases such as the AMA Masterfile [23].

Datasets from 2001 to 2017 were merged into a single joint UPIN-NPI dataset file with total healthcare providers by Zone Improvement Plan (ZIP) codes. A ZIP code to Federal Information Processing Standards (FIPS) crosswalk file was used to create a final sum of total healthcare providers in each county. Some years of data for UPIN and/or NPI directories were missing or inaccessible: 2003, 2008, and 2014. The year 2005 was skipped as it was the year of Hurricane Katrina, as was 2012 for Hurricane Sandy. Last, the year 2006 was dropped for Hurricane Katrina because of a large temporary decrease in population across affected counties, causing a false increase in rates of providers.

Counties affected by Hurricane Katrina and Sandy were identified using Federal Emergency Management Agency disaster declarations (S1 Table) [24]. Affected counties were

limited to one state per storm (Louisiana for Hurricane Katrina and New Jersey for Hurricane Sandy) in order to allow for closer matching of control counties. For each calendar year, we created county healthcare provider to population (per 10,000 adult capita) ratios for each provider type.

## Selection of unaffected locales

For comparison purposes, control counties were randomly selected from U.S. counties unaffected by the respective hurricane and outside of the affected state, selecting counties based on *a priori* specified characteristics: population size and demographics, median county-level household income, number of hospitals, county-level population growth rate and median age. Among these randomly selected counties, counties were then further restricted to counties that (1) were not statistically different from affected counties in socio-economic and demographic characteristics, and (2) did not experience a federally declared disaster of similar scope during the study period. Counties were also excluded that had missing values for at least one of the four types of healthcare providers between 2001 to 2010. Counties that experienced a sizeable influx of Katrina migrants up to a year after the hurricane occurred were also excluded.

## County population measures

We obtained population estimates and county-level sociodemographic measures from U.S. Census estimates for each of the aforementioned years, including county-level estimates of median household income, age, race/ethnicity and sex. The American Hospital Association annual survey and the Area Health Resource File were used to obtain healthcare indicators at the county-level.

## Statistical analyses

For each hurricane we compared the differences pre- versus post-storm by examining the association between county-level healthcare provider supply in affected and control counties. For visual comparison, we display the change in provider to population ratios two years before and five after storms.

A difference-in-differences analysis was conducted in which we compared the difference between the mean county rate of healthcare providers (per 10,000 adult capita) before and after a hurricane for affected and unaffected counties to estimate the effect of exposure to a hurricane on provider availability [18]. We estimated effects based on the change in provider to population ratio in the short-term, by comparing the year before versus two years after the hurricanes, and in a longer period, by comparing the year before to the fifth year after the hurricanes. Difference in trends examined the difference over time between the two slopes of county ratios of providers, testing whether the change between the pre-to-post slope (using years 2001 to 2010 for Hurricane Katrina and 2008 to 2016 for Hurricane Sandy) for the control counties was statistically different from the change between the pre-to-post slope for the disaster counties. An assumption of DID analyses is that there are no significant change between groups relative to one another prior to the change being measured [25]. Parallel trends over time were accounted for in our analysis, using a visual check and a failure to reject the null hypothesis when comparing pre-period means and slopes. We assume that in the absence of the hurricane, the control and treatment groups would continue to have similar trends in provider rates into the post-period. County population growth rates were also controlled for in our analyses through control county case selection.

Selection of appropriate controls (i.e., unaffected counties) is important. Therefore, for Hurricane Katrina we also performed a sensitivity analysis by comparing affected counties to control counties in the state of Louisiana. Considering the entire state of Louisiana was affected to some degree by Hurricane Katrina, the rationale for this additional analysis was to evaluate the change in provider availability within the state of Louisiana, alongside our primary analyses (S2 Table). Analyses were conducted using Stata statistical software (version 15.1; Stata-Corp, College Station, TX). All models used ordinary least squares regression with robust standard errors. For all models, we ran a test of heteroskedasticity and a test of variance inflation factor. All hypothesis tests were two-sided with the critical alpha level at 0.05.

## Results

### Populations affected by Hurricanes Katrina and Sandy

The total population affected by Hurricane Katrina decreased after the storm (total population in affected counties decreased by 18.9 thousand adults), whereas Hurricane Sandy affected counties actually experienced population growth over time (Table 1). Concurrently, the total population among unaffected counties steadily increased over the same time frame for both hurricanes.

County-level characteristics differed in meaningful ways among affected counties of Hurricanes Katrina versus Sandy (Table 1). For instance, the percentage of individuals that were African-American in Hurricane Katrina counties (29.5%) was close to double that of Hurricane Sandy affected populations (14.6%). The median household income in Hurricane Katrina counties was significantly lower than Sandy ($45,800 for Katina versus $65,000 for Sandy). Overall unaffected (i.e., control) counties were similar in characteristics to affected counties for Sandy but several differences were noted among Katrina unaffected counties.

**Table 1. Characteristics of affected and unaffected counties included in study.**

| County characteristic | Hurricane Katrina | | | Hurricane Sandy | | |
|---|---|---|---|---|---|---|
| | Affected by Hurricane (n = 16) | Unaffected by Hurricane (n = 36) | p-value | Affected by Hurricane (n = 8) | Unaffected by Hurricane (n = 21) | p-value |
| Mean county population before hurricane, in thousands | 122.1 | 199.0 | 0.01* | 546.9 | 467.3 | 0.17 |
| Mean county population after hurricane, mean in thousands | 114.8 | 225.2 | 0.01* | 557.4 | 475.8 | 0.15 |
| Population Growth Rate[a], mean (SD) | 0.002 (0.03) | 0.015 (0.01) | 0.06 | 0.003 (0.01) | 0.005 (0.00) | 0.52 |
| Median age, mean (SD)[†] | 36.4 (2.2) | 37.0 (3.4) | 0.38 | 39.6 (4.1) | 38.9 (1.3) | 0.99 |
| Percent female, mean (SD)[†] | 50.3 (1.8) | 50.9 (1.2) | 0.40 | 51.2 (0.7) | 51.2 (0.7) | 0.94 |
| Percent black race/ethnicity, mean (SD)[†] | 29.5 (16.6) | 25.6 (17.7) | 0.37 | 14.6 (12.3) | 11.6 (8.0) | 0.65 |
| Median household income in thousands, mean (SD)[†] | 45.8 (9) | 44 (10) | 0.27 | 65.0 (12.5) | 62.0 (8.5) | 0.68 |
| Total Number of Hospitals, median (IQR)[†] | 2.5 (1, 6) | 2 (1, 5) | 0.95 | 6 (5.5, 8.5) | 5 (5, 8) | 0.18 |

Abbreviations: IQR, interquartile range; SD, standard deviation.

Mann-Whitney test used to compare medians and two sample t-test used to compare means, where p-value compares counties affected by hurricane and a purposefully selected control group of counties unaffected hurricane.

[a] Population growth rate look at change in population for 2001–2010 for Hurricane Katrina and 2010–2017 for Hurricane Sandy.

[†] County characteristics use 2010 census estimates for Hurricane Katrina and for Hurricane Sandy.

*: Statistically significant p-value to the 0.05 level.

## Availability of healthcare providers following Hurricane Katrina

When comparing short-term (2004 vs. 2007) pre- versus post-hurricane county-level health-care provider ratios, primary care physicians decreased by 3.59 providers per 10,000 (95% CI -6.5, -0.7) from pre-hurricane levels. Medical specialists decreased by 5.9 providers per 10,000 (95%CI -11.3, -0.5) and surgeons decreased by 2.1 providers per 10,000 (95% CI -3.8, -0.37) from pre-hurricane levels. Only nurse practitioners did not see an appreciative change (-0.45 provider per 10,000, 95% CI -1.5, 0.6). No significant associations were found when examining the difference in trends over time for difference in difference representation (Tables 2 and 3, and Fig 1).

When comparing long-term (2004 vs. 2010) pre- versus post-hurricane county-level health-care provider ratios, primary care physicians decreased by 4.4 providers per 10,000 (95% CI -7.4, -1.4). Medical specialists decreased by 7.3 providers per 10,000 (95%CI -13, -1.7) and sur-geons decreased by 2.4 providers per 10,000 (95% CI -4.2, -0.53). When we added additional control variables to control for confounding, the magnitude of the variable of interest did not change appreciably for both short-term and long-term changes.

As a sensitivity analysis we evaluated the effects of provider to population ratios among control counties within the state of Louisiana. Across all four provider types, no significant change in provider population ratios of healthcare providers were observed; primary care phy-sicians, medical specialists, surgeons, and nurse practitioners. Similarly, no significant differ-ences in trends were observed. (S2 Table).

## Availability of healthcare providers following Hurricane Sandy

For Hurricane Sandy, little change in availability was observed. When comparing the short-term changes (2011 vs. 2013), healthcare providers per population ratios decreased. Among primary care physicians, the provider per population ratio decreased by 0.9 (95%CI -3.8, -2.0) from a pre-hurricane levels. Medical specialists also had a small decrease, at 1.21 (95%CI -6.5, 4.07), as did nurse practitioners (-0.24, 95%CI -2.6, 2.2) and surgeons (-0.21, 95%CI -1.9, 1.5). However, none of these changes were statistically significant. When comparing long-term changes (2011 vs. 2017), the difference-in-differences changes for providers were small and not statistically significant. No significant changes were found when examining the difference in trends over time. See Tables 2 and 3, and Fig 2.

## Discussion

This study evaluated the association between changes in county-level healthcare provider sup-ply after Hurricanes Katrina in 2005 in Louisiana and Hurricane Sandy in 2012 in New Jersey, using a difference-in-differences analysis. These two historic and devastating hurricanes are test cases to evaluate long-term population level effects on two quite different disaster-affected communities in terms of sociodemographic characteristics. Studying changes in healthcare provider availability can inform long-term community recovery after disaster.

In this study, we found evidence over time of county-level availability decreasing among several types of healthcare providers after Hurricane Katrina including primary care physi-cians, medical specialists and surgeons. No significant changes were seen in affected commu-nities after Hurricane Sandy. This is likely attributable to the differences in the way the two storms impacted communities, but also in the sociodemographic differences between the two settings, including racial and economic disparities. For example, the median household income for Hurricane Katrina affected counties at $45,800 was less than the national average ($50,233) for the study time period, while Hurricane Sandy affected counties at $65,000 were well above the national average ($56,516).

**Table 2. Short-term changes in provider ratios by county among counties impacted by Katrina (2004 vs. 2007) and Hurricanes Sandy (2011 vs. 2013) versus matched controls.**

| | Hurricane Katrina | | | | Hurricane Sandy | | | |
|---|---|---|---|---|---|---|---|---|
| | Pre(SD) | Post (SD) | Change (95% CI) | | Pre (SD) | Post (SD) | Change (95% CI) | |
| | | | No Covariates | With Control Covariates | | | No Covariates | With Control Covariates |
| **Primary Care Physicians** | | | | | | | | |
| *Difference-in-differences* | | | | | | | | |
| Disaster Counties[***] | 6.33 (3.17) | 5.80 (4.56) | -3.61** (-7, -0.2) | -3.59** (-6.5, -0.7) | 9.28 (1.89) | 10.27 (2.06) | -0.84 (-3.7, -2.0) | -0.9 (-3.8, -2.0) |
| Control Counties | 6.54 (2.44) | 9.63 (5.55) | | | 9.21 (3.20) | 11.03 (3.51) | | |
| *Difference-in-trends* | | | | | | | | |
| Disaster Counties | -0.23 | -0.03 | -0.34 (-1.9, 1.2) | -0.35 (-1.6, 0.9) | -0.26 | -0.27 | -0.18 (-2.2, 1.8) | -0.26 (-2.2, 1.7) |
| Control Counties | -0.28 | 0.26 | | | -0.49 | -0.32 | | |
| **Nurse Practitioners** | | | | | | | | |
| *Difference-in-differences* | | | | | | | | |
| Disaster Counties | 0.82 (0.66) | 1.48 (0.87) | -0.45 (-1.5, 0.6) | -0.45 (-1.5, 0.6) | 2.74 (1.22) | 3.37 (1.34) | -0.29 (-2.5, 1.9) | -0.24 (-2.6, 2.2) |
| Control Counties | 2.12 (2.03) | 3.24 (1.89) | | | 4.41 (2.73) | 5.33 (3.11) | | |
| *Difference-in-trends* | | | | | | | | |
| Disaster Counties | -0.085 | 0.298 | 0.03 (-0.5, 0.5) | 0.02 (-1.6, 1.6) | 0.13 | 0.28 | 0.02 (-1.5, 1.5) | 0.02 (-1.6, 1.6) |
| Control Counties | 0.20 | 0.384 | | | 0.16 | 0.28 | | |
| **Medical Specialists** | | | | | | | | |
| *Difference-in-differences* | | | | | | | | |
| Disaster Counties | 12.3 (8.25) | 8.5 (9.84) | -5.3 (-13.3, 2.6) | -5.9** (-11.3, -0.5) | 13.65 (4.63) | 15.31 (4.87) | -1.11 (-7.0, 4.9) | -1.21 (-6.5, 4.07) |
| Control Counties | 14.3 (8.26) | 15.9 (11.9) | | | 13 (5.60) | 15.73 (6.66) | | |
| *Difference-in-trends* | | | | | | | | |
| Disaster Counties | 0.001 | -0.04 | -0.0008 (-3.5, 3.5) | -0.2 (-2.5, 2.1) | -0.54 | -0.63 | -0.13 (-4.3, 4.0) | -0.32 (-3.8, 3.1) |
| Control Counties | 0.277 | 0.23 | | | -0.82 | -0.79 | | |
| **Surgeons** | | | | | | | | |
| *Difference-in-differences* | | | | | | | | |
| Disaster Counties | 3.97 (2.5) | 2.07 (2.2) | -1.98* (-3.13, -0.83) | -2.1** (-3.8, -0.37) | 3.41 (1.29) | 3.85 (1.45) | -0.16 (-1.9, 1.5) | -0.21 (-1.9, 1.5) |
| Control Counties | 4.66 (2.6) | 4.74 (3.7) | | | 3.61 (1.62) | 4.21 (1.76) | | |
| *Difference-in-trends* | | | | | | | | |
| Disaster Counties | -0.046 | 0.08 | -0.001 (-1, 1) | 0.06 (-0.8, 0.7) | -0.19 | -0.18 | 0.03 (-1.1, 1.2) | 0.03 (-1.1, 1.1) |
| Control Counties | -0.021 | 0.10 | | | -0.14 | -0.16 | | |

Note: Provider rates are healthcare providers per 10,000 individuals in a county. Standard errors were heteroskedasticity robust. Control covariates included total population, race and ethnicity, median household income, sex, and total number of hospitals.

Difference in differences time periods were 2004 vs 2007 for Hurricane Katrina and 2011 vs 2013 for Hurricane Sandy.

Differences in trends time periods were 2001–2004 and 2007–2010 for Hurricane Katrina and 2009–2011 to 2013–2015 for Hurricane Sandy.

VIF was under 10 for all difference-in-differences models.

*p<0.1

**p<0.05.

*** County is used as a synonym for Parish.

**Table 3. Long-term changes in provider ratios by county among counties impacted by Katrina (2004 vs. 2010) and Hurricanes Sandy (2011 vs. 2017) versus matched controls.**

| | Hurricane Katrina | | | | Hurricane Sandy | | | |
|---|---|---|---|---|---|---|---|---|
| | Pre (SD) | Post (SD) | Change (95% CI) | | Pre (SD) | Post (SD) | Change (95% CI) | |
| | | | No Covariates | With Control Covariates | | | No Covariates | With Control Covariates |
| **Primary Care Physicians** | | | | | | | | |
| Difference-in-differences | | | | | | | | |
| **Disaster Counties** | 6.33 (3.17) | 5.72 (3.48) | -4.44** (-7.6, -1.2) | -4.4** (-7.4, -1.4) | 9.28 (1.89) | 10.03 (2.07) | -1.08 (-4.0, -1.8) | -0.93 (-4, -2.2) |
| **Control Counties** | 6.54 (2.44) | 10.37 (6.24) | | | 9.21 (3.20) | 11.03 (3.78) | | |
| **Nurse Practitioners** | | | | | | | | |
| Difference-in-differences | | | | | | | | |
| **Disaster Counties** | 0.82 (0.66) | 2.39 (1.11) | -0.67 (-1.9, 0.6) | -0.84 (-2.2, 0.5) | 2.74 (1.22) | 4.93 (1.58) | -0.68 (-3.1, 1.7) | -0.34 (-2.9, 2.2) |
| **Control Counties** | 2.12 (2.03) | 4.36 (2.41) | | | 4.41 (2.73) | 7.23 (3.48) | | |
| **Medical Specialists** | | | | | | | | |
| Difference-in-differences | | | | | | | | |
| **Disaster Counties** | 12.3 (8.25) | 8.4 (9.03) | -6.2 (-13.9, 1.6) | -7.3** (-13, -1.7) | 13.65 (4.63) | 14.58 (5.10) | -1.13 (-7.2, 5.0) | -0.9 (-6.2, 4.4) |
| **Control Counties** | 14.3 (8.26) | 16.6 (12.2) | | | 13 (5.60) | 15.03 (6.77) | | |
| **Surgeons** | | | | | | | | |
| Difference-in-differences | | | | | | | | |
| **Disaster Counties** | 3.97 (2.5) | 2.29 (2.4) | -2.1* (-4.4, 0.24) | -2.4** (-4.2, -0.53) | 3.41 (1.29) | 3.61 (1.451) | -0.33 (-2.1, 1.4) | -0.28 (-2, 1.4) |
| **Control Counties** | 4.66 (2.6) | 5.04 (3.9) | | | 3.61 (1.62) | 4.14 (1.83) | | |

Note: Provider rates are healthcare providers per 10,000 individuals in a county. Difference in differences time periods were 2004 vs 2010 for Hurricane Katrina and 2011 vs 2017 for Hurricane Sandy. Standard errors were heteroskedasticity robust. Control covariates included total population, race/ethnicity, median household income, sex, and total number of hospitals. VIF was under 10 for all difference-in-differences models.

*p<0.1

**p<0.05.

An unprecedented migration from the New Orleans area occurred after Hurricane Katrina, where many thousands left the affected areas. In Louisiana, lower income populations stayed, while higher income populations, including higher-earning healthcare providers, left [17]. Counties affected by Hurricane Katrina already faced socioeconomic disparities (see Table 1); the impact of the storm and its resultant years of recovery undoubtedly exacerbated this.

Availability of medical specialists and primary care physicians were most impacted by Hurricane Katrina. Medical specialists have a broad definition in CMS, and the inclusion of many different types of providers and specialties further boosts our study results. Ratios of nurse practitioners grew considerably over both Katrina and Sandy study time periods, likely due to an increase in volume of nurse practitioners nationwide [26], resulting in difficulty assessing the impact of the hurricanes on NP supply. Primary care physicians had a significant decrease after Hurricane Katrina. This finding is especially critical given that the state of Louisiana continues to have inadequate numbers of primary care physicians to meet the current demand [27], alongside, the demand for primary care physicians outpaced the supply nationwide [28].

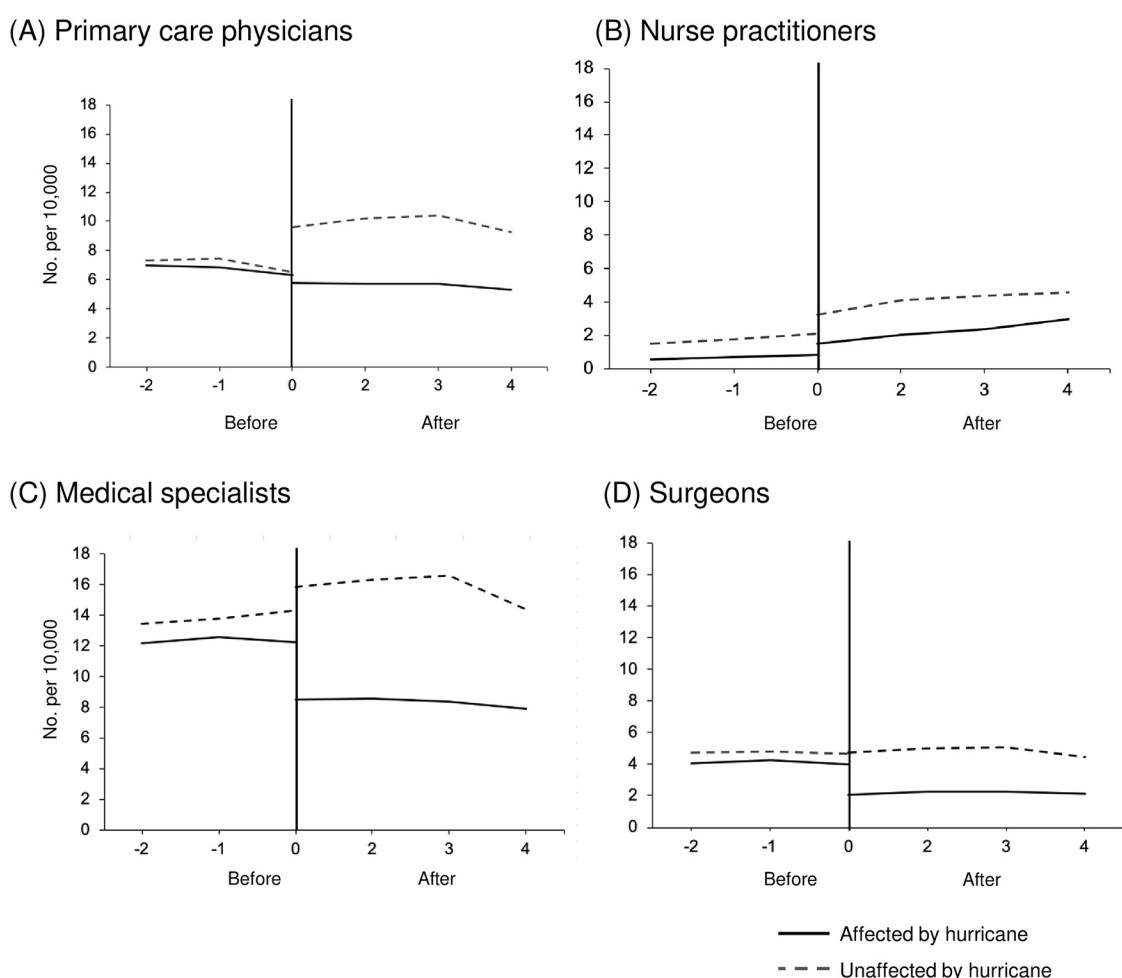

**Fig 1. Provider to population ratio (No. per 10,000) before versus after Hurricane Katrina.** (A) primary care physicians, (B) nurse practitioners, (C) medical specialists, and (D) surgeons. The x-axis consists of study time periods that go from -2 to 4. In the pre-period, time -2, -1, and 0 corresponds with 2001, 2002, and 2004, respectively. In the post-period, time 0, 2, 3, and 4 corresponds with 2007, 2009, 2010, and 2011, respectively.

Attracting primary care providers (including nurse practitioners) to areas with long-term post-disaster recovery needs is a difficult prospect that policymakers have struggled to address.

While our analyses demonstrated statistically significant differences in overall availability, we did not detect meaningful differences in trends (i.e., the slopes of availability before versus after storms). Finally, in our sensitivity analysis, using controls within the state of Louisiana, no significant findings were observed in either analysis. This implies that changes in provider supply were observed state-wide in Louisiana where virtually no parishes (the term parish is used by the State of Louisiana, in place of the equivalent term, county) were spared differential impacts from the hurricane. In fact, non-affected parishes had overall lower rates of all providers compared to affected parishes both before and after the disaster.

This study was not without limitations. First, the NPPES was designed for administrative purposes, not for tracking the health professional workforce. Comparisons to other data sources suggest that this data source provides a reasonably accurate picture of the aggregate supply of providers across the geographic regions studied [23]. Second, there are limitations associated with NPI data, where certain types of providers are likely better represented in the

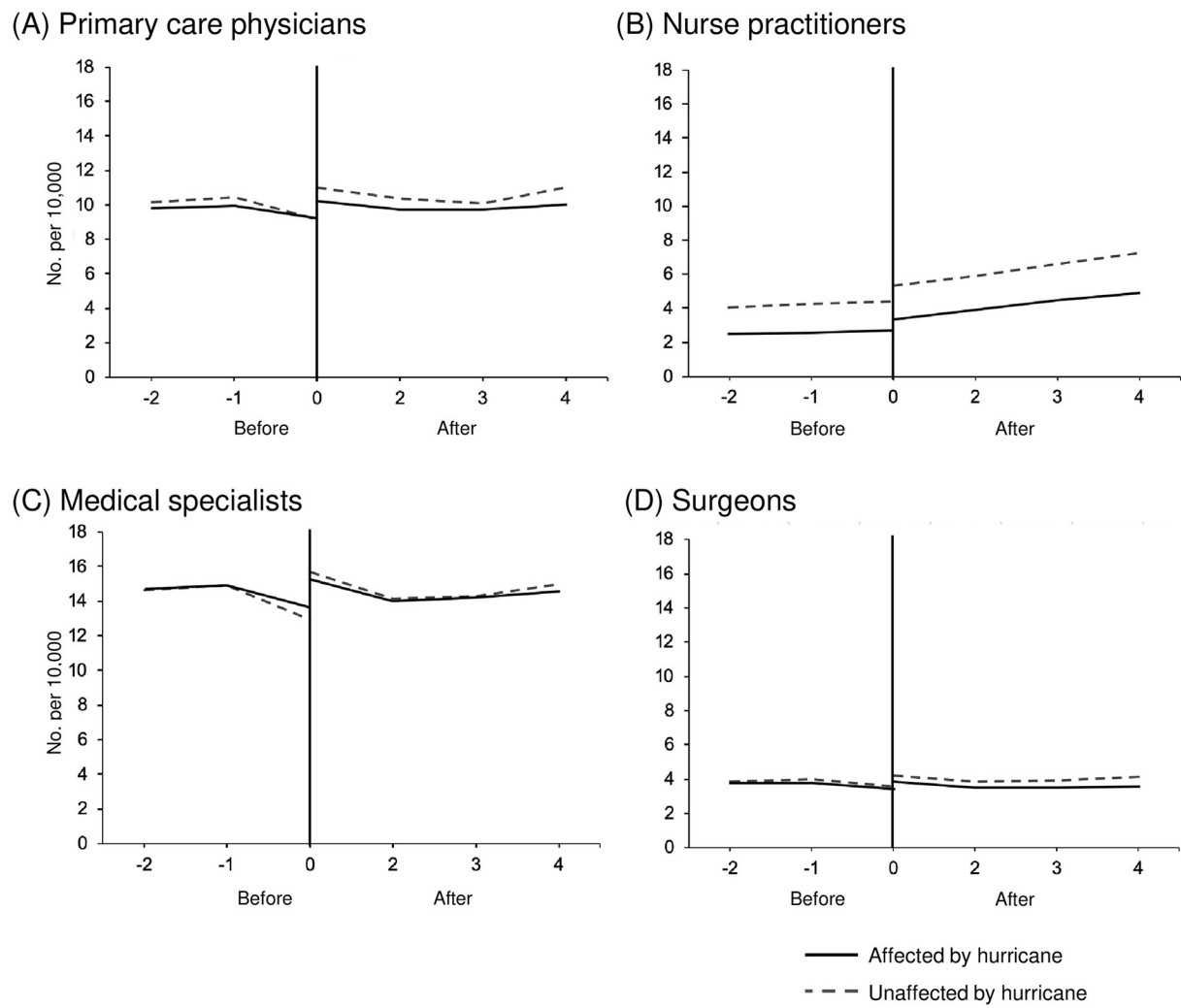

**Fig 2. Provider to population ratio (No. per 10,000) before versus after Hurricane Sandy.** (A) primary care physicians, (B) nurse practitioners, (C) medical specialists, and (D) surgeons. The x-axis consists of study time periods that go from -2 to 4. In the pre-period, time -2, -1, and 0 corresponds with 2009, 2010, and 2011, respectively. In the post-period, time 0, 2, 3, and 4 corresponds with 2013, 2015, 2016, and 2017, respectively.

data than those who may bill under another provider's NPI or organization NPI, such as nurse practitioners. Providers who are not actively practicing can still have an NPI number. Third, the switch from UPIN to NPI data occurred in 2007; and, it most affected NPs and PCPs. Thus, the Hurricane Katrina DiD analysis for PCPs and NPs could have picked up this change. Finally, criteria for selection of control counties was based on literature and policy review, after considering other alternatives such as propensity matching [29].

Availability of healthcare providers has direct impacts on the health of affected communities, making this issue an important aspect of disaster recovery. Limited healthcare provider availability contributes to adverse individual and population level health effects, where these effects are compounded in communities recovering from disaster. These needs are known to be greater in communities that are sociodemographically disadvantaged, making the lack of medical specialists and primary care physicians especially concerning. Efforts to address the primary care provider shortage occurred after Hurricane Katrina, for example through a

Primary Care Access and Stabilization Grant to the Louisiana Department of Health and Hospitals from the U.S. Department of Health and Human Services [30]. This program was intended to support the restoration of restore access to health care in communities affected by Hurricane Katrina.

Community recovery from disaster has no specific endpoint, and most consider both hurricane-affected regions as still in the recovery process today. Creative methods are needed to encourage primary care providers to return to, or to relocate to, communities that may not be attractive as they struggle to recover. Health workforce policy, such as that from the Health Resources and Services Administration (HRSA), could work alongside disaster and emergency management policymakers to include healthcare provider considerations in long-term recovery planning. The growing population of nurse practitioners may be one solution to address primary care needs. However, they cannot be the only solution, as medical specialists and surgeons remain critically needed.

## Conclusion

The study found that effects of hurricanes on healthcare availability were largely contingent on the population-level characteristics of affected counties, where race/ethnicity and economic characteristics are key contributors. Large-scale disasters continue to have devastating effects across the United States. Populations that are most in need of healthcare, such as aging populations, those with disabilities or chronic conditions, and children, are the most impacted by the community-level disruption from these events, making the need for regular healthcare even more critical in communities that have protracted recovery periods. Long term trends in county-level healthcare provider supply after Hurricanes Sandy and Katrina suggest the need for policy level efforts to ensure stability in provision of care to disaster-affected communities.

## Supporting information

**S1 Table. Case and control counties.**
(DOCX)

**S2 Table. Changes in provider ratios by county among counties impacted by Hurricane Katrina and in-state Louisiana controls.**
(DOCX)

## Author Contributions

**Conceptualization:** Sue Anne Bell, Matthew A. Davis.

**Data curation:** Sue Anne Bell.

**Formal analysis:** Sue Anne Bell, Katarzyna Klasa.

**Funding acquisition:** Sue Anne Bell.

**Methodology:** Sue Anne Bell, Theodore J. Iwashyna, Edward C. Norton, Matthew A. Davis.

**Project administration:** Sue Anne Bell.

**Supervision:** Sue Anne Bell, Theodore J. Iwashyna, Edward C. Norton, Matthew A. Davis.

**Validation:** Matthew A. Davis.

**Writing – original draft:** Sue Anne Bell, Katarzyna Klasa, Matthew A. Davis.

**Writing – review & editing:** Sue Anne Bell, Katarzyna Klasa, Theodore J. Iwashyna, Edward C. Norton, Matthew A. Davis.

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
