## [Decision Letter · Decision Letter 0]

19 Aug 2020

PONE-D-20-17250

Long-Term Healthcare Provider Availability Following Large-scale Hurricanes: A Difference-in-Differences Study

PLOS ONE

Dear Dr. Bell,

Thank you for submitting your manuscript to PLOS ONE. After careful consideration, we feel that it has merit but does not fully meet PLOS ONE’s publication criteria as it currently stands. Therefore, we invite you to submit a revised version of the manuscript that addresses the points raised during the review process.

I concur with the methodological issues raised by Reviewer 1. Control variables should be clearly distinguished from treatment variables. Direct effects of treatment variables both with and without control variables must be reported. How do treatment effects change with the addition of control variables, and why? Standard statistical tests on multi-collinearity, misspecification, auto-corellation etc. need to be reported. The discussion section needs to elaborate model limitations  that might arise due to biases induced by model specification, multi-collinearity etc. 

We look forward to receiving your revised manuscript.

Kind regards,

Asim Zia, Ph.D.

Academic Editor

PLOS ONE

Additional Editor Comments:

I concur with the methodological issues raised by Reviewer 1. Control variables should be clearly distinguished from treatment variables. Direct effects of treatment variables both with and without control variables must be reported. How do treatment effects change with the addition of control variables, and why? Standard statistical tests on multi-collinearity, misspecification, auto-corellation etc. need to be reported. The discussion section needs to elaborate model limitations that might arise due to biases induced by model specification, multi-collinearity etc.

2. Please include captions for your Supporting Information files at the end of your manuscript, and update any in-text citations to match accordingly. Please see our Supporting Information guidelines for more information: http://journals.plos.org/plosone/s/supporting-information

Reviewers' comments:

Reviewer's Responses to Questions

**Comments to the Author**

1. Is the manuscript technically sound, and do the data support the conclusions?

Reviewer #1: Yes

2. Has the statistical analysis been performed appropriately and rigorously? 

Reviewer #1: Yes

3. Have the authors made all data underlying the findings in their manuscript fully available?

Reviewer #1: Yes

4. Is the manuscript presented in an intelligible fashion and written in standard English?

Reviewer #1: Yes

5. Review Comments to the Author

Reviewer #1: This study examined the availability of healthcare providers following two US natural disasters: Hurricanes Sandy and Katrina. The authors used publicly available provider and county-level data. The difference-in-difference approach used is appropriate given the study design and question. The finding that counties affected by Katrina had more substantial changes in healthcare provider availability compared with counties affected by Sandy raises important questions as to how to tailor and target interventions aiming to restore access to healthcare following natural disasters. I have several questions or comments, enumerated below.

Did any of the affected counties have missing/0 providers during the study period for any of the provider categories?

How did you identify counties that had a “sizeable influx of Katrina migrants”? This was listed as an exclusion criteria for control counties.

The methods stats that control counties were randomly selected and determined not to be statistically significantly different in sociodemographics than affected counties, but in Table 1 there is a significant difference in county population between control and affected counties before the hurricane Katrina.

Why is the term parish used sometimes instead of county? Is that a meaningful distinction?

The authors hypothesize that the decrease in providers after Katrina may be due to providers migrating. A potential driver could be hospital closures, which then force providers to look elsewhere for work. They have number of hospitals by county in Table 1 – does that number change following Katrina?

Is there a threshold that is used to determine whether an area is officially determined to have a provider shortage? If so, do the declines following Katrina meet this threshold? A decline on its own does not necessarily indicate a shortage.

Given the similar results for the short term and long term DID analyses, it seems the impact of disasters on healthcare provider availability is relatively permanent/stable. Perhaps more discussion on this is warranted. Was there any targeted intervention attempted after Katrina to recruit healthcare providers back to the area?

6. PLOS authors have the option to publish the peer review history of their article (what does this mean?). If published, this will include your full peer review and any attached files.

Reviewer #1: **Yes: **Erika Moen

---

## [Author Response · Author response to Decision Letter 0]

6 Oct 2020

A table of revisions that provides detail on all reviewer comments has been uploaded.

---

## [Decision Letter · Decision Letter 1]

10 Nov 2020

Long-Term Healthcare Provider Availability Following Large-scale Hurricanes: A Difference-in-Differences Study

PONE-D-20-17250R1

Dear Dr. Bell,

We’re pleased to inform you that your manuscript has been judged scientifically suitable for publication and will be formally accepted for publication once it meets all outstanding technical requirements.

Kind regards,

Asim Zia, Ph.D.

Academic Editor

PLOS ONE

Additional Editor Comments (optional):

Reviewers' comments:

Reviewer's Responses to Questions

**Comments to the Author**

1. If the authors have adequately addressed your comments raised in a previous round of review and you feel that this manuscript is now acceptable for publication, you may indicate that here to bypass the “Comments to the Author” section, enter your conflict of interest statement in the “Confidential to Editor” section, and submit your "Accept" recommendation.

Reviewer #1: All comments have been addressed

2. Is the manuscript technically sound, and do the data support the conclusions?

Reviewer #1: Yes

3. Has the statistical analysis been performed appropriately and rigorously? 

Reviewer #1: Yes

4. Have the authors made all data underlying the findings in their manuscript fully available?

Reviewer #1: Yes

5. Is the manuscript presented in an intelligible fashion and written in standard English?

Reviewer #1: Yes

6. Review Comments to the Author

Reviewer #1: (No Response)

7. PLOS authors have the option to publish the peer review history of their article (what does this mean?). If published, this will include your full peer review and any attached files.

Reviewer #1: No

---

## [Editor Report · Acceptance letter]

13 Nov 2020

PONE-D-20-17250R1 

Long-Term Healthcare Provider Availability Following Large-scale Hurricanes: A Difference-in-Differences Study 

Dear Dr. Bell:

I'm pleased to inform you that your manuscript has been deemed suitable for publication in PLOS ONE. Congratulations! Your manuscript is now with our production department. 

Kind regards, 

on behalf of

Professor Asim Zia 

Academic Editor

PLOS ONE